# Hierarchical and Heterogeneous Porosity Construction and Nitrogen Doping Enabling Flexible Carbon Nanofiber Anodes with High Performance for Lithium-Ion Batteries

**DOI:** 10.3390/ma15134387

**Published:** 2022-06-21

**Authors:** Jun Liu, Yuan Liu, Jiaqi Wang, Xiaohu Wang, Xuelei Li, Jingshun Liu, Ding Nan, Junhui Dong

**Affiliations:** 1Inner Mongolia Key Laboratory of Graphite and Graphene for Energy Storage and Coating, School of Materials Science and Engineering, Inner Mongolia University of Technology, Hohhot 010051, China; clxylj@163.com (J.L.); ly2547068790@163.com (Y.L.); wjq024466341811@163.com (J.W.); wxh20220208@163.com (X.W.); jingshun_liu@163.com (J.L.); jhdong@imut.edu.cn (J.D.); 2Rising Graphite Applied Technology Research Institute, Chinese Graphite Industrial Park—Xinghe, Ulanqab 013650, China; 3Collaborative Innovation Center of Non-ferrous Metal Materials and Processing Technology Co-Constructed by the Province and Ministry, Inner Mongolia Autonomous Region, Inner Mongolia University of Technology, Hohhot 010051, China; 4College of Chemistry and Chemical Engineering, Inner Mongolia University, West University, Street 235, Hohhot 010021, China; nd@imu.edu.cn; 5Inner Mongolia Enterprise Key Laboratory of High Voltage and Insulation Technology, Inner Mongolia Power Research Institute Branch, Inner Mongolia Power (Group) Co., Ltd., Hohhot 010020, China

**Keywords:** lithium-ion battery, flexible anode, hierarchical and heterogeneous porosity structure, N-doping, high performances

## Abstract

With the rapid development of flexible electronic devices, flexible lithium-ion batteries are widely considered due to their potential for high energy density and long life. Anode materials, as one of the key materials of lithium-ion batteries, need to have good flexibility, an excellent specific discharge capacity, and fast charge–discharge characteristics. Carbon fibers are feasible as candidate flexible anode materials. However, their low specific discharge capacity restricts their further application. Based on this, N-doped carbon nanofiber anodes with microporous, mesoporous, and macroporous structures are prepared in this paper. The hierarchical and heterogeneous porosity structure can increase the active sites of the anode material and facilitate the transport of ions, and N-doping can improve the conductivity. Moreover, the N-doped flexible carbon nanofiber with a porous structure can be directly used as the anode for lithium-ion batteries without adding an adhesive. It has a high first reversible capacity of 1108.9 mAh g^−1^, a stable cycle ability (954.3 mAh g^−1^ after 100 cycles), and excellent rate performance. This work provides a new strategy for the development of flexible anodes with high performance.

## 1. Introduction

Lithium-ion batteries have become the most widely used energy storage devices due to their long service life, high safety, low self-discharge, rapid charge and discharge, high power/energy densities, and environmental friendliness [1,2]. Nowadays, the application of lithium-ion batteries can be seen everywhere, such as in the aerospace industry, satellite communication, mobile phones, watches, and other portable electronic devices. In recent years, flexible and portable electronic devices have gradually entered the market, including flexible display screens, wearable electronic devices, flexible electronic medical devices, etc. The flexible lithium-ion battery is one of the important parts of flexible electronic devices, which has the advantages of a light weight, small volume, large energy storage capacity, long service life, wide temperature adaptation range, and high specific energy compared to other flexible energy devices [3]. Flexible electrode materials are the main factor determining the electrochemical performance of flexible lithium-ion batteries [4,5]. Therefore, they have become a research hotspot for scholars to develop flexible electrode materials with excellent electrochemical properties.

Carbon fibers are used extensively as flexible lithium-ion anode materials due to their good conductivity and mechanical strength [6]. However, carbon fiber anodes face the problems of poor flexibility, low reversible capacity, and insufficient rate performance when used directly. To improve their electrochemical performance, an effective measure is to fabricate holes of different sizes in the carbon fibers or dope the carbon fibers with some active substances. Zhang et al. reported that micropores could provide a larger surface area, which is beneficial to the distribution of reaction sites [7]. Mesopores were reported to greatly enhance the rate capability and facilitate the transport of ions [8]. Additionally, Zhu et al. reported that macropores could accommodate great volume variation during cycling [9]. At present, N-doping is also widely considered due to the advantages of low cost, stable combination of carbon and nitrogen, and improved performance [10,11]. However, it is difficult for a single mending strategy to fully satisfy the requirements of high electrochemical performances. Therefore, it may be an effective strategy to coordinate various modification methods for improving the performance of carbon nanofiber anodes.

Based on the above analysis, we produced a flexible N-doped carbon nanofiber anode with microporous, mesoporous, and macroporous structures using gas–electric co-spinning technology in this work. Compared to commonly used electrospinning, this technology can not only greatly improve the production efficiency but can also reduce the power consumption caused by a high-voltage power supply. In addition, this technology has a series of advantages such as device simplicity, easy operation, and low cost. The hierarchical and heterogeneous porosity structure can increase the number of reactive active sites of lithium ions and facilitate the transport of ions to effectively improve the reversible capacity of the anode, and N-doping can improve the conductivity. The N-doped carbon nanofiber anode with hierarchical and heterogeneous porosity structure can be directly used as the anode of lithium-ion batteries without adding adhesives and any substrate. Moreover, it has a high first reversible capacity of 1108.9 mAh g^−1^, capacity retention of 86.1% after 100 cycles, and excellent rate performance, which is much higher than the results reported for other carbon fiber anodes for lithium-ion batteries. This work provides a new way to manufacture flexible carbon nanofiber anodes with high performance for lithium-ion batteries.

## 2. Experimental

### 2.1. Materials

The reagents used in this paper included polyacrylonitrile (PAN) (solid, molecular weight 150,000; Macklin), graphene (solid, 5 μm; Jiangnan Graphene Research Institute, Changzhou, China), N, N-dimethylformamide (DMF) (liquid, analytical purity; Macklin), and melamine (C_3_H_6_N_6_) (solid, analytically pure; Macklin), which were of analytical purity to be used directly.

### 2.2. Preparation of NH_3_-Activated Carbon Nanofibers

Gas–electric co-spinning technology combines electrostatic spinning technology and gas spinning technology. First, 2 g polyacrylonitrile (PAN) were added to N, N-dimethylformamide (DMF) and stirred for 10 h at 70 °C in a water bath to form a 10% PAN/DMF solution. The prepared precursor solution was put into a 50-milliliter syringe which had a coaxial stainless steel needle with an inner diameter of 1.11 mm and an outer diameter of 1.49 mm. An airflow nozzle was connected to the needle, and then the syringe was placed on the propeller. The collecting plate was 15 × 20 cm wire mesh. The distance between the needle and the collecting plate was controlled at 10~15 cm, and a high-voltage generator was used to connect the syringe needle and the steel wire mesh. Then, the prepared precursor solution was continuously spun on the collecting plate for 1 h to obtain a fiber cloth. In the process of gas–electric co-spinning, the feed rate was 4 mL h^−1^, the voltage was 5 kV, and the airflow was 10 psi. For pre-oxidation, the obtained fiber cloth was pre-oxidized in a blast drying oven at 280 °C at a heating rate of 2 °C min^−1^ for 6 h. The purpose of pre-oxidation was to make PAN undergo three chemical reactions, namely cyclization, dehydrogenation, and oxidation, so as to make the carbon nanofiber cloth more stable before carbonization. For carbonization, the pre-oxidized fiber cloth was placed into a high-temperature tubular furnace, and the temperature was raised from room temperature to 950 °C at a rate of 3 °C min^−1^ under the protection of a high-purity nitrogen atmosphere for 1 h. Then, after carbonization, the high-purity nitrogen was replaced by ammonia and kept for 0–40 min for activation. Finally, the ammonia was changed into high-purity nitrogen gas and cooled to room temperature to obtain NH_3_-activated carbon nanofibers. In the process of preparing the above materials, the NH_3_-activated carbon nanofibers were named CNFs-0NH_3_, CNFs-10NH_3_, CNFs-20NH_3_, CNFs-30NH_3_, and CNFs-40NH_3_ according to the activation times of 0, 10, 20, 30, and 40 min, respectively.

### 2.3. Preparation of N-Doped Carbon Nanofibers

The preparation process of N-doped carbon nanofibers with the hierarchical and heterogeneous porosity structure is shown in Figure 1. First, 0.1 g graphene was dissolved in 20 g DMF and sonicated for 20 min. Then, 2 g PAN and 1 g C_3_H_6_N_6_ were added to the above solution and stirred for 10 h at 70 °C in a water bath to form a uniform spinning solution. Then, the prepared precursor solution was spun continuously for 1 h to obtain a nanofiber cloth by the gas–electric co-spinning device. The conditions for the gas–electric co-spinning and subsequent pre-oxidation were the same as above. The activation time of ammonia was 30 min. In the process of preparing the above materials, according to the mass ratios of PAN to C_3_H_6_N_6_ of 2:0, 2:1, 2:2, and 2:3, the final products were named CNFs-0N, CNFs-1N, CNFs-2N, and CNFs-3N, respectively.

### 2.4. Materials Characterization

The sample morphology was characterized using scanning electron microscopy (SEM, HITACHI-SU8220, HITACHI, Tokyo, Japan), and the corresponding element mapping on the surface of the materials was analyzed using an energy-dispersive spectrometer (EDS). The composition, content, and chemical valence of surface elements were studied by X-ray photoelectron spectroscopy (XPS, ESCALAB 250Xi, Thermo Fisher, Waltham, MA, USA). The pore size distribution of samples was measured at 77.2 K by the Brunauer–Emmett–Teller method (BET, BELSORP-mini II, BEL Japan Inc., Osaka, Japan). Before the pore size distribution test, the samples needed to be degassed for 8 h at 200 °C under an N_2_ atmosphere to remove the adsorbed water molecules and low volatile content in the material.

### 2.5. Electrochemical Measurements

The prepared anode sheet of lithium-ion cells was a flexible carbon nanofiber which did not need the addition of binders or to be coated on copper foil as with traditional electrodes. The prepared N-doped carbon nanofiber anodes with the hierarchical and heterogeneous porosity structure after being sliced can be used directly as an anode. The electrode loading for the cells was ~1.2 mg cm^−2^, and the diameter of this electrode was ~0.9 cm. A metal lithium sheet was used as the counter electrode, the separator was porous polypropylene, and the electrolyte was an EC:DEC:EMC = 1:1:1 (*v*/*v*) solvent with added 20% fluoroethylene carbonate (FEC) and lithium hexafluorophosphate (LiPF_6_). The volume of the electrolyte used in each of the cells was about 0.4 mL. The electrochemical experiments were carried out using CR2032 cells at room temperature. The blue electric system (Land CT2001A, Blue Power Company) was set at constant current charge–discharge between the voltage range of 0.01 and 3 V. The rate performance of cells was tested at different current densities (50, 100, 200, 500, and 1000 mA g^−1^). A Princeton (PMC1000A) electrochemical workstation was used to test the cyclic voltammetry (CV) in the voltage range of 0.01~3 V with a scanning rate of 0.01 mV s^−1^. Additionally, electrochemical impedance spectroscopy (EIS) tests were carried out between 0.1 Hz and 100 kHz with an amplitude of 5 mV.

## 3. Results and Discussion

Figure 2 show the microscopic morphology of carbon nanofibers before activation and after activation for 30 min by ammonia. It can be seen from the images that the carbon nanofiber has a good fiber shape, and its diameter is about 300 nm. The SEM images in Figure 2a show that some particulate structures emerge on the surface of the carbon nanofibers before activation, which is caused by pre-oxidation and carbonization. In Figure 2b, the surface of the carbon nanofibers after activation for 30 min by ammonia not only has no small agglomerations but also shows the appearance of abundant pores, including the microporous, mesoporous, and macroporous structures produced by ammonia activation. The hierarchical and heterogeneous porosity structure is instrumental in increasing the active sites and facilitates the transport of ions of flexible carbon anode materials [7,8,9].

To analyze the types and states of surface elements of the carbon nanofibers after different ammonia activation times, all NH_3_-activated carbon nanofibers were tested by XPS. The full spectrum in Figure 3a shows that these carbon nanofibers after ammonia activation contained three elements: C, N, and O; the corresponding three orbits C_1s_, N_1s_, and O_1s,_ were located at 285, 400, and 533 eV, respectively. The C/N atomic ratios of CNFs-10NH_3_, CNFs-20NH_3_, CNFs-30NH_3_, and CNFs-40NH_3_ were 4.57%, 5.64%, 5.76%, and 6.76%, respectively. The results show that ammonia activation can not only make pores but can also dope N, and with the increase in ammonia activation time, the N content in the carbon nanofibers also increased. Furthermore, Figure 3b–e show the N_1s_ spectra of the carbon nanofibers after different ammonia activation times. It can clearly be seen that all samples have two peaks of pyridinic nitrogen (N-6) and pyrrolic/pyridone nitrogen (N-5), which are located near 398.5 and 400.1 eV, respectively. It is reported that the existence of pyridine nitrogen and pyrrole nitrogen in carbon nanofibers can increase the conductivity [12,13]. Simultaneously, pyridine nitrogen and pyrrole nitrogen have a strong ability to adsorb lithium, which can improve the lithium storage performance of carbon nanofibers.

The effect of the different ammonia activation times on the electrochemical properties of the carbon nanofiber anode was tested in lithium-ion batteries. Figure 4a show the cycle performance curves of the CNFs-0NH_3_, CNFs-10NH_3_, CNFs-20NH_3_, CNFs-30NH_3_, and CNFs-40NH_3_ anodes at the current density of 50 mA g^−1^ from 0.01 to 3 V. The digital picture in Figure 4a shows the anode sheet activated by ammonia, which is flexible and foldable and can be used directly as a flexible lithium-ion anode without the addition of binders. It should be noted that the capacity vs. cycle number curves (and the CE curves as well) exhibit a lot of fluctuations (or wavy nature), especially at the slow rate of 50 mA g^−1^. This is mainly because the temperature of the laboratory was affected by the ambient temperature, which led to the temperature fluctuation of the test battery near the room temperature. However, in the whole test process, this performance change trend was basically correct, and the data can explain the performance of the materials. With the increase in the ammonia activation time, the initial capacity of the carbon nanofiber anode gradually increased. After 80 cycles, the reversible capacity of the CNFs-0NH_3_, CNFs-10NH_3_, CNFs-20NH_3_, CNFs-30NH_3_, and CNFs-40NH_3_ anodes was 406.3, 424.4, 440.2, 728.1, and 689.2 mAh g^−1^, respectively, indicating that the CNFs-30NH_3_ anode had the best cycle stability. Figure 4b show the rate performance curves of the CNFs-0NH_3_, CNFs-10NH_3_, CNFs-20NH_3_, CNFs-30NH_3_, and CNFs-40NH_3_ anodes. Similarly, with the increase in the ammonia activation time from the CNFs-0NH_3_, CNFs-10NH_3_, and CNFs-20NH_3_ anodes to the CNFs-30NH_3_ anode, the rate performance of the carbon nanofiber anodes also gradually increased. The CNFs-30NH_3_ anode showed the strongest rate performance, with an average specific capacity of 818.4, 666.5, 623.8, 508.9, and 452.4 mAh g^−1^ at the current density of 50, 100, 200, 500, and 1000 mA g^−1^, respectively. When the current density returned to 50 mA g^−1^, the specific capacity of the CNFs-30NH_3_ anode could still reach 736.7 mAh g^−1^. However, with the further increase in the ammonia activation time, the rate performance of the CNFs-40NH_3_ anode was attenuated. These results illustrate that a moderate ammonia activation time is beneficial to improving the electrochemical properties of carbon nanofibers, as excessive activation may make the anode sheet brittle due to the presence of too many holes, resulting in poor rate performance.

Based on the analysis of the above results, ammonia activation can make carbon nanofibers obtain a hierarchical and heterogeneous porosity structure, which is instrumental in promoting electrochemical properties. In addition, it is worth noting that a small amount of nitrogen was doped in the carbon nanofibers in the process of manufacturing the microporous, mesoporous, and macroporous structures. Therefore, to further explore the effect of N-doping on the electrochemical performance of carbon nanofiber anodes, we chose CNFs-30NH_3_ with the best performance as the N-doping object to reveal the influence mechanism of nitrogen on carbon nanofibers with the hierarchical and heterogeneous porosity structure and to further improve its electrochemical performance.

Figure 5a–d show the micromorphology of the hierarchical and heterogeneous porosity structure of carbon nanofibers with different N-doping contents. The images show that all N-doped porous carbon nanofibers have a good nanofiber shape, with a diameter of about 300 nm. With the increase in the N-doping content, there was little difference in the micromorphology of the carbon nanofibers. However, there are many cross-linking phenomena in the CNFs-3N anode. This is because when the added content of melamine is 3 g, the spinnability of the nanofiber becomes poor, and the spinning rate decreases due to the high viscosity of the solution, so the CNFs-3N anode will be locally uneven after pre-oxidation, carbonization, and activation. The SEM image of CNFs-2N and the corresponding EDS mapping of elements are shown in Figure 5e. The C and N elements are evenly distributed, indicating that N was effectively doped in the nanofibers. Figure 5f show the nitrogen adsorption–desorption curves. The hysteresis loop of each sample appears at P/P_0_ > 0.4, belonging to type IV isotherms. This indicates that all samples contain mesopores. Furthermore, the corresponding pore size distribution curves of CNFs-0N, CNFs-1N, CNFs-2N, and CNFs-3N in Figure 5g indicate that all samples have micropores (pore size less than 2 nm), mesopores (pore size between 2 and 50 nm), and macropores (pore size greater than 50 nm), which meets the expectation of preparing N-doped carbon nanofibers with a hierarchical and heterogeneous porosity structure.

To further analyze the types and states of surface elements on the porous carbon nanofibers with different nitrogen contents, CNFs-0N, CNFs-1N, CNFs-2N, and CNFs-3N were tested by XPS. The full spectra in Figure 6a show that the prepared carbon nanofibers with different nitrogen contents contained three elements: C, N, and O; the corresponding three orbits, C_1s_, N_1s_, and O_1s_, were located at 285, 400, and 533 eV, respectively. The C/N atomic ratios of CNFs-0N, CNFs-1N, CNFs-2N, and CNFs-3N were 5.8%, 6.6%, 8.4%, and 9.9%, respectively, obtained by XPS data analysis. The C/N atomic ratios of all samples were higher than those of CNFs-30NH_3_, indicating that the amount of nitrogen was effectively increased by adding melamine during preparation. Figure 6b–e show the N_1s_ spectra of CNFs-0N, CNFs-1N, CNFs-2N, and CNFs-3N. All samples have two peaks of pyridinic nitrogen (N-6) at 398.5 eV and pyrrolic/pyridone nitrogen (N-5) at 400.1 eV. These results explain that nitrogen was effectively doped into CNFs-30NH_3_, and the chemical bond formed by doping melamine is consistent with that formed by ammonia activation.

Figure 7a show the cycle performance curves of the CNFs-0N, CNFs-1N, CNFs-2N, and CNFs-3N anodes at 50 mA g^−1^ from 0.01 to 3 V in lithium-ion batteries. After 100 cycles, the reversible capacity of each anode changed little. In particular, the CNFs-2N anode had a high initial capacity of 1108.9 mAh g^−1^ and still maintained a high reversible capacity of 954.3 mAh g^−1^ after 100 cycles, which was the highest capacity among the four anodes. However, the initial Coulombic efficiency of the CNFs-0N, CNFs-1N, CNFs-2N, and CNFs-3N anodes was only 49.72%, 50.24%, 46.29%, and 45.71%, respectively. The low initial Coulombic efficiency is due to the fact that after ammonia activation, the materials developed a number of micropores, mesopores, and macroporous structures, providing many reactive sites and forming a high irreversible capacity with the formation of a large area of SEI film during the first charge and discharge process. After the first charge and discharge, the Coulomb efficiency of each anode was stable at basically more than 90%, which is attributed to the formation of stable SEI film.

Figure 7b show the rate performance curves of the CNFs-0N, CNFs-1N, CNFs-2N, and CNFs-3N anodes in lithium-ion batteries at different current densities. The CNFs-2N anode still exhibited excellent specific capacities of 1047.2, 796.9, 709.6, 619.9, and 558.3 mAh g^−1^ at 50, 100, 200, 500, and 1000 mA g^−1^, respectively. When the current density changed to 50 mA g^−1^ again, the specific capacity of the CNFs-2N anode was higher than 921.2 mAh g^−1^, indicating a good rate performance. This is because N-doping and the hierarchical and heterogeneous porosity structure can improve the conductivity, increase the reactive sites, and improve the Li-ion transport rate. To the best of our knowledge, the outstanding electrochemical performance of the CNFs-2N anode is much better than that of other reported carbon fiber anodes for lithium-ion batteries (Table 1).

Figure 7c show the constant-current charge–discharge curves of the CNFs-2N anode during the 1st, 10th, and 50th cycles in the voltage range of 0.01~3 V. In the first cycle, the discharge and charge capacities were observed to be 2395 and 1108.9 mAh g^−1^, respectively, corresponding to 46.3% of the initial Coulombic efficiency. For the 10th and 50th cycles, the charge capacity was 879 and 906 mAh g^−1^, respectively, and the discharge capacity was 929 and 963 mAh g^−1^, respectively. Compared with the first cycle, the decrease in capacity was due to the formation of SEI film and the irreversible lithiation of carbon in the initial discharge process [14]. To further estimate the lithium storage performance of the CNFs-2N anode, CV curves of the first five cycles were obtained from 0.01 to 1.5 V at a scan rate of 0.1 mV s^−1^, as shown in Figure 7d. During the first lithiation, the generation of the SEI film and the decomposition of the electrolyte led to a broad irreversible peak at about 0.55 V, which disappeared in the following cycles. For the anodic peak at 0.51 V and the reduction peak at 0.21 V, the current intensity gradually increased with the increase in the number of cycles, which corresponds to the activation process of more active materials reacting with lithium ions [34].

To further explore the effect of N-doping on the performance of lithium-ion batteries, the EIS of the CNFs-0N and CNFs-2N anodes was tested. Figure 7e show the Nyquist plots of the CNFs-0N and CNFs-2N anodes in lithium-ion batteries. The curve consists of a semicircle in the high-frequency region and a slash in the low-frequency region. The semicircle in the high-frequency region represents the charge transfer impedance. The oblique line in the low-frequency region is related to lithium-ion diffusion [35]. The semicircle in the high-frequency region of the CNFs-2N anode is larger than that of the CNFs-0N anode—that is, the charge transfer impedance increases after N-doping. The increase in the charge transfer impedance after N-doping may be due to the formation of a larger SEI film after forming pores, which makes CNFs-2N possess a larger specific surface area. However, the ideal pore size distribution can shorten the ion diffusion path and provide more active sites. Moreover, the porous structure can accommodate the volume strain and inhibit the volume expansion in the process of Li^+^ insertion/de-insertion to achieve long-term cycle stability. Therefore, the CNFs-2N anode has a high specific capacity, an excellent rate performance, and cycle stability. In addition, compared with CNFs-0N, CNFs-2N has a higher slope in the low-frequency region. The larger the slope is, the lower the lithium-ion diffusion resistance is. This indicates that CNFs-2N has high conductivity after N-doping.

## 4. Conclusions

In this work, flexible N-doped carbon nanofiber anodes with a hierarchical and heterogeneous porosity structure were synthesized, which can be directly used as the anode of lithium-ion batteries without adding adhesives. SEM images and nitrogen adsorption–desorption tests confirmed the existence of microporous, mesoporous, and macroporous structures in the carbon nanofibers. EDS and XPS spectra proved that the nitrogen element was successfully doped in the carbon nanofibers. The hierarchical and heterogeneous porosity structure increased the active sites of the anode materials, improved the ion transport rate, and inhibited the volume expansion in the process of Li^+^ insertion/de-insertion. The N-doping improved the conductivity of the carbon nanofibers. As expected, the prepared CNFs-2N anode had a high initial reversible specific capacity of 1108.9 mAh g^−1^ and excellent capacity retention and rate performance in lithium-ion batteries. This work provides a new way to develop high-performance flexible anode materials for lithium-ion batteries.

## Figures and Tables

**Figure 1 materials-15-04387-f001:**
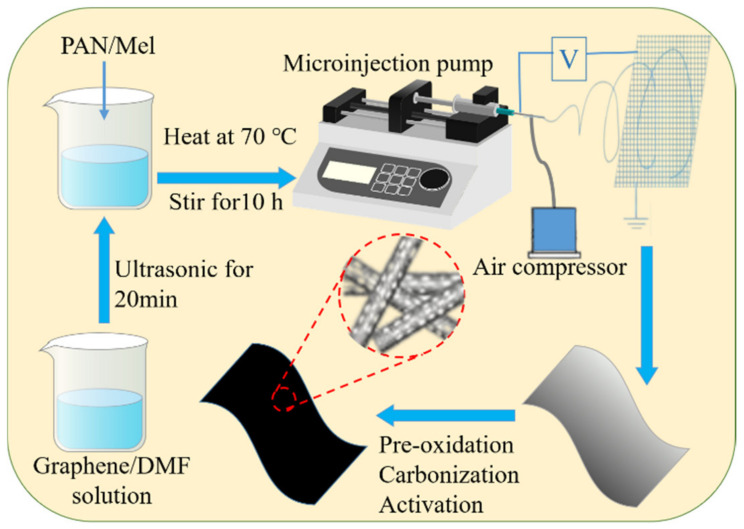
Flow chart of the preparation of N-doped carbon nanofiber anodes with the hierarchical and heterogeneous porosity structure.

**Figure 2 materials-15-04387-f002:**
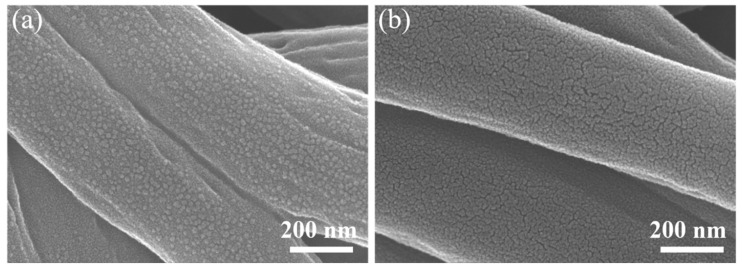
SEM images of carbon nanofibers (**a**) before activation and (**b**) after activation for 30 min by ammonia.

**Figure 3 materials-15-04387-f003:**
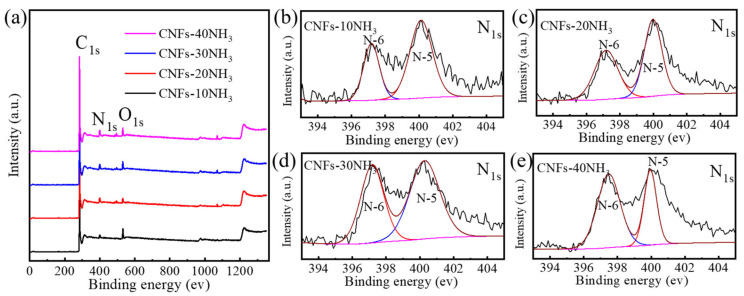
XPS analysis of carbon nanofibers with different ammonia activation times: (**a**) Full spectra of CNFs-10NH_3_, CNFs-20NH_3_, CNFs-30NH_3_, and CNFs-40NH_3_. N_1s_ spectra of (**b**) CNFs-10NH_3_, (**c**) CNFs-20NH_3_, (**d**) CNFs-30NH_3_, and (**e**) CNFs-40NH_3_.

**Figure 4 materials-15-04387-f004:**
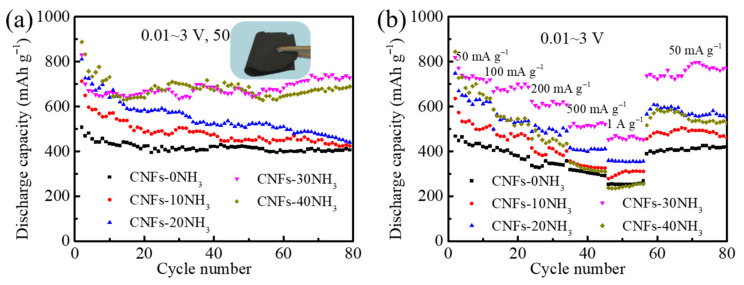
(**a**) Cycle performance curves and (**b**) rate performance curves of CNFs-0NH_3_, CNFs-10NH_3_, CNFs-20NH_3_, CNFs-30NH_3_, and CNFs-40NH_3_ anodes. The digital picture in (**a**) is the anode sheet activated by ammonia.

**Figure 5 materials-15-04387-f005:**
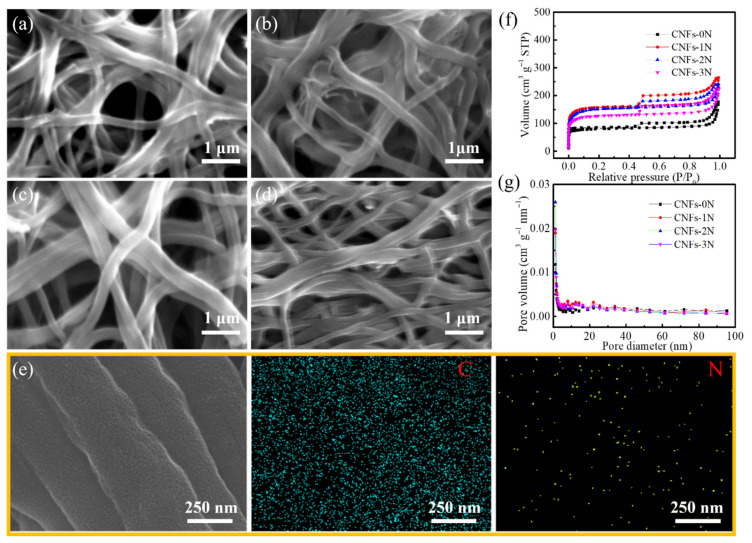
SEM images of porous carbon nanofibers with different nitrogen doping contents: (**a**) CNFs-0N, (**b**) CNFs-1N, (**c**) CNFs-2N, and (**d**) CNFs-3N. (**e**) SEM image and the corresponding EDS mapping of C and N elements in partial CNFs-2N. (**f**) Nitrogen adsorption–desorption curves and (**g**) pore size distribution curves of CNFs-0N, CNFs-1N, CNFs-2N, and CNFs-3N.

**Figure 6 materials-15-04387-f006:**
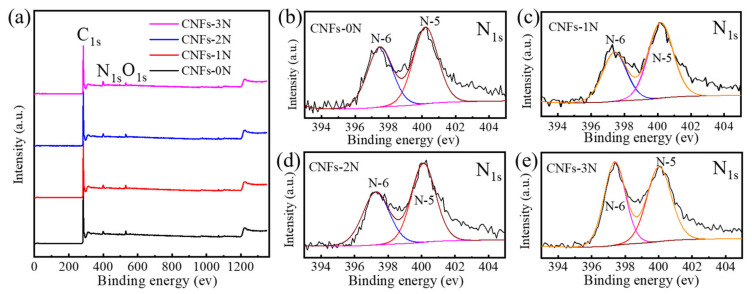
XPS spectra of porous carbon nanofibers with different nitrogen contents: (**a**) Full spectra of CNFs-0N, CNFs-1N, CNFs-2N, and CNFs-3N. Spectra of N_1s_: (**b**) CNFs-0N, (**c**) CNFs-1N, (**d**) CNFs-2N, and (**e**) CNFs-3N.

**Figure 7 materials-15-04387-f007:**
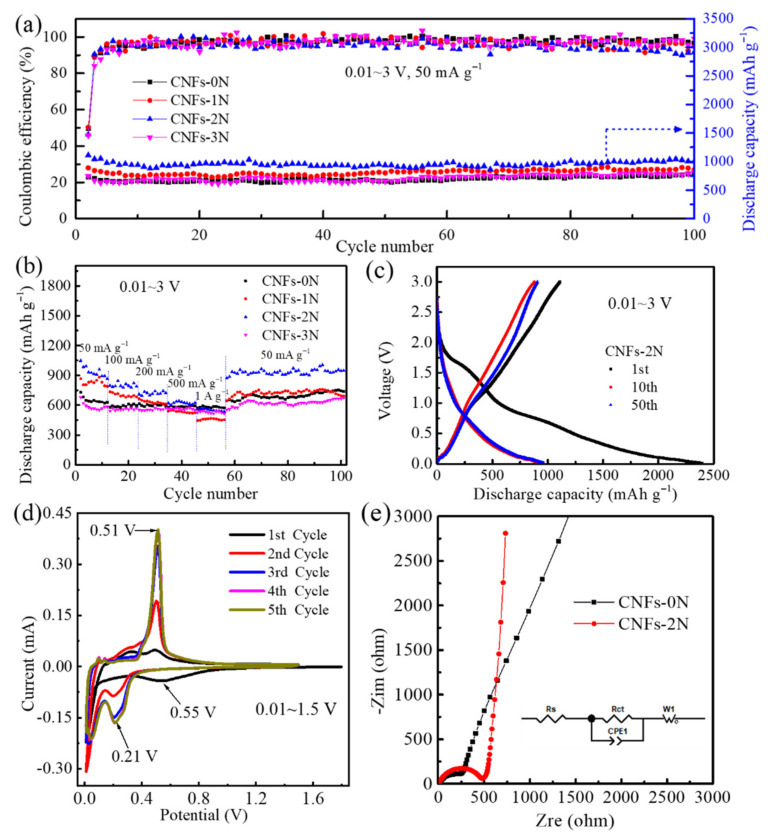
(**a**) Cycle performance curves and (**b**) rate performance curves of CNFs-0N, CNFs-1N, CNFs-2N, and CNFs-3N anodes. (**c**) The charge–discharge curves of the CNFs-2N anode, (**d**) CV curves of the CNFs-2N anode, and (**e**) Nyquist plots of CNFs-0N and CNFs-2N anodes in lithium-ion batteries. The insertion diagram in (**e**) is the equivalent circuit.

**Table 1 materials-15-04387-t001:** Electrochemical performances of flexible carbon fiber anodes for lithium-ion batteries reported elsewhere and in this work.

Anodes	Mass Loading (mg cm^−2^)	First Reversible Capacity (mAh g^−1^)	Cycle Performance (mAh g^−1^)	Rate Performance (mAh g^−1^)	Reference
V_2_O_3_/MCCNFs	1.5~2.5	790.6 (0.1 A g^−1^)	487.7 (5 A g^−1^, 5000 cycles)	456.8 (5 A g^−1^)	[7]
Sn@C@CNF	2.0	891.2 (0.1 A g^−1^)	610.8 (0.2 A g^−1^, 180 cycles)	305.1 (2 A g^−1^)	[9]
SnS/CNFs	/	898 (0.05 A g^−1^)	548 (0.5 A g^−1^, 500 cycles)	206 (4 A g^−1^)	[14]
γ-Fe_2_O_3_/C films	1.0	923.97 (0.2 A g^−1^)	1088 (0.2 A g^−1^, 300 cycles)	380 (5 A g^−1^)	[15]
am-Fe_2_O_3_/rGO/CNFs	1.5~2.0	825 (0.1 A g^−1^)	739 (1 A g^−1^, 400 cycles)	570 (2 A g^−1^)	[16]
In_2_O_3_@CF	1.4	510 (0.1 A g^−1^)	435 (0.1 A g^−1^, 500 cycles)	190 (1.5 A g^−1^)	[17]
MnSe@C-700	1.6	614.6 (0.1 A g^−1^)	684 (0.1 A g^−1^, 100 cycles)	/	[18]
NCNFs	7.64	752.3 (0.05 A g^−1^)	411.9 (0.1 A g^−1^, 160 cycles)	148.8 (2 A g^−1^)	[19]
CNF@SnO_2_	1.77~3.54	793 (0.5 A g^−1^)	485 (0.1 A g^−1^, 850 cycles)	359 (4 A g^−1^)	[20]
G/Si@CFs	0.65~1	1036 (0.1 A g^−1^)	896.8 (0.1 A g^−1^, 200 cycles)	543 (1 A g^−1^)	[21]
C/CuO/rGO	1.30~1.95	550 (0.1 A g^−1^)	400 (1 A g^−1^, 600 cycles)	300 (2 A g^−1^)	[22]
FeCo@NCNFs-600	1.77~2.65	736.3 (0.1 A g^−1^)	566.5 (0.1 A g^−1^, 100 cycles)	130 (2 A g^−1^)	[23]
SnO_2_/TiO_2_@CNFs	/	1061.2 (0.1 A g^−1^)	729.6 (0.1 A g^−1^, 150 cycles)	206.2 (3 A g^−1^)	[24]
MoO_2_/C	85.7	752.5 (0.2 A g^−1^)	450 (2 A g^−1^, 500 cycles)	432 (2 A g^−1^)	[25]
FCNF-3/4	1.0	775 (0.2 A g^−1^)	630 (0.2 A g^−1^, 100 cycles)	250 (5 A g^−1^)	[26]
Fe_3_O_4_/NCNFs	1.33	686 (0.1 A g^−1^)	522 (0.1 A g^−1^, 200 cycles)	407 (5 A g^−1^)	[27]
Fe_2_O_3_/SnO_x_/CNF	/	797 (0.1 A g^−1^)	756 (0.1 A g^−1^, 55 cycles)	540 (1 A g^−1^)	[28]
V_2_O_3_/CNF	/	415.3 (0.2 A g^−1^)	420 (0.2 A g^−1^, 100 cycles)	80 (10 A g^−1^)	[29]
ZnSe@CNFs-2.5	0.8~1.2	737.5 (0.1 A g^−1^)	426.1 (5 A g^−1^, 3000 cycles)	547.6 (5 A g^−1^)	[30]
SiOC/C fibers-NH	0.8~1.5	518 (0.2 A g^−1^)	595 (0.2 A g^−1^, 200 cycles)	195 (4 A g^−1^)	[31]
10-SnO_2_@CNFs/CNT	1.5~2.5	500.9 (0.1 A g^−1^)	460.3 (0.1 A g^−1^, 200 cycles)	222.2 (3.2 A g^−1^)	[32]
γ-Fe_2_O_3_@CNFs	2.0	1065 (0.5 A g^−1^)	430 (6 A g^−1^, 1000 cycles)	222 (60 A g^−1^)	[33]
CNFs-2N	1.2	1108.9 (0.05 A g^−1^)	954.3 (0.05 A g^−1^, 100 cycles)	549.7 (1 A g^−1^)	This work

## Data Availability

Not applicable.

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
