# Peer review of "Hierarchical and Heterogeneous Porosity Construction and Nitrogen Doping Enabling Flexible Carbon Nanofiber Anodes with High Performance for Lithium-Ion Batteries"

_materials, 2022, doi:10.3390/ma15134387_

Round 1

Reviewer 1 Report

The manuscript, “Micro-/meso-pores construction and nitrogen-doping enabling high-performance carbon nanofiber anodes for lithium-ion batteries” reports the effect of nitrogen doping on carbon nanofibers and their influence on the electrochemical performance when used as an anode in Li-ion batteries. The results appear interesting, and the reported capacity values seem quite high. The manuscript may be of interest to the Materials community, however, there are a few important concerns in the manuscript that need to be addressed or clarified before it can be accepted for publication. They are listed as follows:

  1. In Figure 4a-b, and Figure 7b, the capacity vs. cycle number curves (and the CE curves as well) exhibit a lot of fluctuations (or) wavy nature, especially at the slow rate of 50 mA/g. Why is this the case? Can the authors provide an explanation or hypothesis in the manuscript?
  2. What is the electrode loading for the cells used? Could the authors specify this in the Methods/Electrochemical Measurements section? I understand that there is no slurry involved, but it is important to mention the weight of the carbon fibers that go into each cell
  3. How exactly were the carbon nanofiber slices/sheets directly used as anodes? Was any substrate used to support the carbon fiber “slices”?
  4. What was the volume of the electrolyte used in each of the cells? This needs to be specified as well
  5. In Figure, 5e, what was the equivalent circuit used to fit the Nyquist plots? The x-axis and y-axis should ideally be on equal scales to compare the shape of the semi-circles more effectively
  6. The language in the manuscript should be checked thoroughly for grammatic and spelling errors

Reviewer 2 Report

This article studies anodes for lithium-ion batteries made of N-doped carbon nanofibers. The authors propose two different synthetic ways to produce carbon nanofibers, doping them with different nitrogen precursors at several conditions of time and concentration, obtaining performing materials for application as freestanding and flexible anodes. To characterize the materials and the electrochemical performances the authors used suitable techniques and they discussed properly the results.

The manuscript is interesting and overall well written, but In order to publish this work I have some comments and revisions to suggest:

1) In the Introduction part, it would be helpful for the reader to have a deeper comparison with the state of the art and in particular with the electrochemical performance of other carbon nanostructures and / or carbon fibers. This contextualization could also be highlighted in the part of the “Results and discussion”, for example with a table, to highlight the novelty and the quality of the results obtained in this study.

2) In the “Experimental” chapter it is necessary to specify the graphene used and the company from which it was purchased for a correct reproducibility of the work and because with the term graphene you can mean many types of different products.

In general, the synthesis description is not very clear and it should be improved by better describing the steps of oxidation, carbonization and activation.

3) As regards the characterization of the materials, the porosity study part should be clarified. In particular, the authors claim to obtain micro and mesoporous materials and it is not clear why the presence of macro porosity is excluded, which instead would seem evident from SEM analysis. In my opinion, the system has a hierarchical and heterogeneous porosity and the authors should try to discuss and better clarify the role of the different porosity with respect to nitrogen and lithium in the final application.

4) The authors used SEM technique showing the morphology of the materials, especially to highlight the doping with nitrogen. However, a measure of energy-dispersive-spectroscopy (EDS) with a map of N would have been useful in order to have a clear visual evidence of the dopant and its dispersion in the material.

Reviewer 3 Report

The article titled “Micro-/meso-pores construction and nitrogen-doping enabling high-performance carbon nanofiber anodes for lithium-ion batteries” by Liu et al. describes a comprehensive study of the nitrogen-doped carbon nanofibers and the application as flexible anodes for lithium-ion batteries. The experiments were well-designed, and the article is well-organized. This article would provide valuable insights for researchers in lithium-ion batteries. I hereby recommend the publication of this article with the following comments addressed:

1. English language editing is needed to further improve readability of this article. 

2. Since flexible lithium-ion batteries is the main research background presented in the introduction part, I would recommend mentioning “flexible” in the article title.

3. Page 1, “The flexible lithium-ion batteries are an essential part of flexible electronic devices”. This statement needs to be modified. There are other power sources and energy devices for flexible electronics. Please see section 3 of this review paper: https://doi.org/10.1002/advs.202001116. The authors should address the advantages of flexible lithium-ion batteries compared to other flexible energy devices. 

4. Page 2, section 2.2, the authors mentioned in the preparation of the carbon nanofibers that a “gas-electric co-spinning device” was used. It would be good to provide more information about this fabrication method. What is the advantage compared to commonly used electrospinning?

5. Page 4, “Figure 2a shows that many nano-sized particles appear on the surface of carbon nanofibers before activation”. From the SEM images it seems to be some particulate structures on the surface of the carbon nanofibers. Assumedly the structures should be the same materials as the carbon nanofibers? Using “nano-sized particles” is a little confusing here.

6. Page 6, Figure 5f. Does this plot mean the distribution of pore volume vs. different pore size (width)? It would be helpful if the authors could elaborate more on the pore size distribution measurement and provide more details on how to translate the results from Figure 5e to Figure 5f.

7. Page 7, about the interpretation of EIS results, “the semicircle in the high frequency region represents…The slash in the low frequency region is related to …” Please double check. The high frequency semi-circle should be the charge-transfer resistance and the low frequency tail should be the Li ion diffusion.

Round 2

Reviewer 1 Report

The authors have satisfactorily addressed my comments in the revised manuscript. Therefore, I recommend accepting the revised manuscript for publication.